# Diabetic Proteinuria Revisited: Updated Physiologic Perspectives

**DOI:** 10.3390/cells11182917

**Published:** 2022-09-18

**Authors:** Samuel N. Heyman, Itamar Raz, Jamie P. Dwyer, Roni Weinberg Sibony, Julia B. Lewis, Zaid Abassi

**Affiliations:** 1Department of Medicine, Hadassah Hebrew University Hospital, Mt. Scopus, Jerusalem 9765422, Israel; 2Division of Geriatrics, Herzog Hospital, Jerusalem 9765422, Israel; 3Faculty of Medicine, Hebrew University of Jerusalem, Jerusalem 9765422, Israel; 4Diabetes Unit, Department of Endocrinology and Metabolism, Hadassah Medical Center, Jerusalem 9124001, Israel; 5Clinical and Translational Science Institute, University of Utah Health, Salt Lake City, UT 84112, USA; 6Faculty of Medicine, Ben-Gurion University, Beer Sheva 84105, Israel; 7Division of Nephrology and Hypertension, Vanderbilt University Medical Center, Nashville, TN 37232, USA; 8Departments of Medicine and Nephrology, Vanderbilt University Medical Center, Nashville, TN 37011, USA; 9Department of Physiology and Biophysics, Rappaport Faculty of Medicine, Technion-Israel Institute of Technology, Haifa 3200003, Israel; 10Department of Laboratory Medicine, Rambam Health Care Campus, Haifa 3109601, Israel

**Keywords:** diabetes mellitus, diabetic nephropathy, albuminuria, glomerular, tubular, renal reserve, high-protein diet, SGLT-2 inhibitors, ACE inhibitors, angiotensin receptor blockers

## Abstract

Albuminuria, a hallmark of diabetic nephropathy, reflects not only injury and dysfunction of the filtration apparatus, but is also affected by altered glomerular hemodynamics and hyperfiltration, as well as by the inability of renal tubular cells to fully retrieve filtered albumin. Albuminuria further plays a role in the progression of diabetic nephropathy, and the suppression of glomerular albumin leak is a key factor in its prevention. Although microalbuminuria is a classic manifestation of diabetic nephropathy, often progressing to macroalbuminuria or overt proteinuria over time, it does not always precede renal function loss in diabetes. The various components leading to diabetic albuminuria and their associations are herein reviewed, and the physiologic rationale and efficacy of therapeutic interventions that reduce glomerular hyperfiltration and proteinuria are discussed. With these perspectives, we propose that these measures should be initiated early, before microalbuminuria develops, as substantial renal injury may already be present in the absence of proteinuria. We further advocate that the inhibition of the renin–angiotensin axis or of sodium–glucose co-transport likely permits the administration of a normal recommended or even high-protein diet, highly desirable for sarcopenic diabetic patients.

## 1. Introduction

Diabetes is the leading cause of end-stage kidney disease in developed countries, reflecting the overwhelming epidemic of obesity and associated metabolic syndrome [1]. In developing countries, diabetic kidney disease is also on the rise. Albuminuria, evolving from microalbuminuria to nephrotic-range proteinuria, is a clinical hallmark of diabetic nephropathy (DN). It develops in about a third of diabetic patients and is considered an independent risk factor in the progression of DN and for all-cause mortality [2,3]. However, many diabetic patients exhibit progressive deterioration of kidney function without proteinuria [4], a condition termed nonproteinuric diabetic kidney disease [5], underscoring the pathological heterogeneity and dynamic nature of DN [6,7]. As schematically illustrated in Figure 1, the pathogenesis of proteinuria in the diabetic kidney is far more complicated than simply the damaged glomerular filtration barrier (GFB). 

Diabetic nephropathy leads to enhanced albumin leak into the initial urine by (a) glomerular hyperfiltration and increased trans-glomerular pressure, and (b) by structural and functional changes at the filtration barrier, permeating albumin leak into Bauman’s capsule. 

Filtered albumin is reabsorbed mainly by proximal tubular cells. A fraction of albumin binds to neonatal Fc receptor (FcRn), undergoes trans-cytosis and is reclaimed intact into the bloodstream. Another fraction undergoes endocytosis, attached to megalin and cubilin, and goes through proteolysis with the release of amino acids (AA) into the bloodstream. Free fatty acids dissolved in albumin are up taken as well and are subjected to beta oxidation with the generation of acetyl-Co A, but some accumulate in tubular cells with the formation of oval fat bodies and tubular injury, with the induction of inflammation that culminates in chronic and progressive tubulointerstital disease.

Thus, albuminuria traced in the final urine reflects initial glomerular leak minus albumin reclaiming by tubular cells. Albuminuria appears, first as microalbuminuria, when glomerular leak of albumin exceeds maximal tubular capacity of albumin reabsorption. It increases as glomerular leak intensified, and/or as tubular uptake capacity diminishes.

Diabetic nephropathy progressing to CKD reflects both glomerular changes (direct injury as well as wear and tear damage caused by hyperfiltration) and tubulointerstitial disease. High protein intake enhances glomerular hyperfiltration, augmenting proteinuria with consequent tubulointerstitial disease. By contrast, glomerular hyperfiltration and downstream glomerular and tubulointerstitial damage are attenuated by angiotensin II receptor blockers (ARBs) and inhibitors of angiotensin-converting enzyme (ACEi) and sodium–glucose co-transporter 2 (SGLT2i).

The outstanding capacity of the liver to enhance albumin synthesis is likely hampered by renal and systemic inflammation. Thus, hypoalbuminemia develops when renal albumin loss (albuminuria, combined with tubular albumin reclaiming and degradation) surpasses undermined hepatic albumin synthesis. Abbreviations: AA—amino acids; ARB—angiotensin II-receptor blockers; ACEI—angiotensin-converting-enzyme inhibitors; SGLT2I—sodium–glucose cotransporter-2 inhibitors.

In fact, diabetes leads to profound functional and structural changes along the nephron, affecting the glomerili, tubuli and renal interstitium (Figure 2).

This review addresses current concepts regarding the additional role of glomerular hyperfiltration and tubulopathy in the evolution of diabetic proteinuria and nephropathy and addresses recent controversial findings such as discrepancies between renal morphology and the degree of proteinuria. The nature of glomerular acute and protracted hyperfiltration is outlined, attributed to protein intake and to enhanced sodium–glucose co-transport, respectively, and changes related to renal functional reserve and to structural glomerular remodeling, correspondingly. The impact of proteinuria on the progression to CKD is underscored, and the debate regarding dietary protein restriction is revisited. Finally, we advocate for an early initiation of interventions that lower glomerular hyperfiltration, even before the development of microalbuminuria.

Albuminuria is a common feature of diabetic nephropathy (DN). This phenomenon reflects not only damage to the glomerular filtration barrier but is also affected by altered glomerular hemodynamics and hyperfiltration, as well as by the inability of renal tubular cells to fully retrieve filtered albumin. At the glomerular level, hyperfiltration, endothelial dysfunction, thickened basement membrane, podocyte’s effacement/effusion and detachment, glomerular sclerosis and hyalinosis may all increase albumin leak at the glomerular apparatus. However, there are other components that may lead to diabetic albuminuria or non-diabetic nephropathy. For instance, downregulation of cubilin/megalin, tubular inflammation, injury and atrophy and reduced reabsorption of amino acids and proteins by proximal tubules may play a role in the development of albuminuric DN. Moreover, reduced distal tubule reabsorption of albumin, probably by the downregulation of assumedly expressed megalin or cubilin there, may also affect the severity of albuminuria. Likewise, the presence of interstitial inflammation and fibrosis during DM may negatively impact tubular capacity to reabsorb albumin and eventually determine the severity of diabetic proteinuria.

## 2. Pathophysiology

### 2.1. The Leaky Glomerulus—Deformed Filtration Barrier

Albuminuric diabetic patients display glomerulopathy with mesangial sclerosis and arteriolar hyalinosis. These renal histological changes have been reported to appear earlier and are more pronounced in patients with type 1 diabetes mellitus (T1D) compared with those with type 2 diabetes mellitus (T2D) [8]. Microalbuminuria may precede these glomerular gross morphologic changes, reflecting subtle structural microangiopathy that is not readily identified by light microscopy. Diabetic glomerular microangiopathy develops along with retinal and perineuronal microvasculopathy, resulting from glucotoxicity with elevated advanced glycation end-products (AGEs), dysregulated insulinemia and glomerular hyperfiltration with barotrauma, affecting all components of the GFB [9]. Endothelial dysfunction, increased deposition of extracellular matrix with disrupted glomerular basement membrane (GBM), loss of podocyte permselectivity and progressive podocyte damage are all components of this initial GFB injury. The complex nature of diabetic glomerulopathy detailed below is schematically depicted in Figure 3A.

A.Under normal conditions, GFB prevents leakage of high-molecular-weight proteins, including albumin, into Bowman’s space. Destruction of any of the layers of the GFB in the diabetic kidney might result in proteinuric disease. This may be initiated by promoting increased production of reactive oxygen species (ROS), the induction of AGE-induced proinflammatory signaling and increased glomerular capillary pressure and hyperfiltration. Damage to all three components of GFB in the diabetic kidney is evident by endothelial dysfunction (ED), disrupted glomerular basement membrane (GBM) with increased deposition of extracellular matrix, loss of podocyte permselectivity and progressive podocyte damage: (1) ED is characterized by structural and functional damage to the glycocalyx, and its components, especially heparan sulfate (HS), are released and appear in the urine. Indeed, derangement of the glycocalyx plays a pivotal role in the development of albuminuria secondary to enhanced vessel wall permeability with protein leak. (2) One of the milestone features of DN is thickening of GBM along with the development of proteinuria. (3) Similarly, podocyte damage is considered a crucial step in the pathogenesis of proteinuric kidney diseases, including DN. Specifically, effacement and effusion of podocyte foot processes are a hallmark feature of ultrastructural alterations characterizing proteinuric renal disease, including DN.B.Albumin is reabsorbed mainly by proximal tubular cells via two pathways: (1) a fraction of albumin binds to neonatal Fc receptor (FcRn), undergoes trans-cytosis and is reclaimed intact into the bloodstream; (2) a fraction of albumin attaches to megalin and cubilin and undergoes endocytosis and subsequent proteolysis in lysosomes, with the release of amino acids (AA) into the bloodstream. An unknown percentage of filtered albumin is likely reclaimed intact by the paracellular pathway. Free fatty acids (FFA) dissolved in albumin are up taken as well and are subjected to beta oxidation with the generation of acetyl-Co A. A fraction of FFA accumulates in tubular cells with the formation of oval fat bodies and tubular injury. This leads to the induction of inflammation that culminates in chronic and progressive tubulointerstitial disease with tubular atrophy and interstitial fibrosis.

**Endothelial cells (ECs):** Hyperglycemia, insulin resistance and hyperinsulinemia independently cause EC dysfunction by promoting increased production of reactive oxygen species (ROS), activation of protein kinase C (PKC) and the induction of AGE-induced proinflammatory signaling [10]. Lesions of glomerular ECs are common in patients with DN [11], with EC dysfunction progressing to histologic glomerular sclerotic disease [12,13]. Albuminuria is considered a hallmark of generalized vascular endothelial disease [14]. Diabetes is one of the leading causes of structural and functional damage to the glycocalyx [15]. This could contribute to enhanced vessel wall permeability with protein leak, though this possibility has been questioned [16]. Additional indices of glomerular EC damage are inflammation, mitochondrial injury, oxidative stress, aberrant cell signaling, increased EC permeability and endothelial-to-mesenchymal transition (EMT) [17,18]. Increased renal expression of glycocalyx-degrading enzymes such as hyaluronidase and heparanase were observed in diabetic patients, a finding which may contribute to the development of albuminuria [9]. Furthermore, glomerular ECs are in contact with the GBM and exert an interdependent relationship with podocytes and mesangial cells, an interplay via growth factors and cellular signaling. Indeed, damage to ECs impairs podocytes and vice versa [19,20]. For instance, glomerular endothelial injury in experimental diabetes was found to be associated with podocyte loss, mediated by the activation of the endothelin system [21]. Endothelial abnormalities are more closely associated with increased urine albumin excretion than podocyte injury, suggesting that endothelial dysfunction may play a more critical role in GFB alterations in DN than podocyte damage [9]. This notion is further supported by Ryan and Karnovsky [22], who have reported that the endothelial layer is the most effective barrier preventing albumin filtration. Thus, damage to ECs in many clinical settings, including DM, may adversely affect podocyte integrity and function with the development of mesangial proliferation and glomerulosclerosis, along with albuminuria [12,13,17].

**Glomerular basement membrane (GBM):** Lesions involving the GBM are common in many immune and metabolic disorders, including DM [23]. One of the milestone features of DN is thickening of GBM along with the development of proteinuria. Light- and electron-microscopic and immunofluorescence techniques have revealed that GBM thickening induced by diabetes emerges from the overproduction of its components by podocytes and mesangial cells, with a regression of ECs to an embryonic production mode [24,25]. Moreover, hyperglycemia provokes impaired matrix turnover due to inflammatory, oxidative and profibrotic pathways. Interestingly, two variants in GBM matrix production were found to affect diabetic albuminuria. COL4A3: 1-rs34505188 (causing an Arg408His sub situation) was detected in African American diabetics, promoting albuminuria, while 2-rs55703767 (causing an Asp326Tyr substitution), found in Europeans with T1D, was shown to provide nephroprotection with reduced albuminuria [23]. The hypothesis linking a loss of heparan sulphate proteoglycan-related anionic sites in the glomerular basement membrane to diabetic albuminuria has been generally abandoned, based on direct determination of such sites and by the lack of their association with the degree of proteinuria in diabetics [26].

**Podocytes:** These glomerular epithelial cells are important components of the GFB, with their unique slit diaphragm and related interdigitating proteins. These terminally differentiated cells have a limited ability to proliferate. Therefore, podocyte damage is considered a crucial step in the pathogenesis of proteinuric kidney diseases, including DN [11]. Specifically, the effacement of podocyte foot processes is a hallmark feature of ultrastructural alterations characterizing proteinuric renal disease, including DN [27]. The association between the effacement of foot processes and proteinuria has been proven both experimentally and clinically [28,29]. For instance, several studies demonstrated that proteinuric DN may develop because of podocytopathy provoked by high glucose levels, by insulin resistance, and due to inflammatory and toxic insults caused by advanced glycation end-products (AGEs) [30]. In this context, it was reported that hyperglycemia causes mitochondrial overload and exaggerated ROS production, which further destruct glomerular podocytes and promote their apoptosis [31,32].

Most recently, a published cohort study (TRIDENT—Transformative Research in Diabetic Nephropathy) by Palmer et al. showed that glomerular epithelial hypertrophy is part of an early maladaptive compensation to glomerulomegaly observed in patients with diabetes. Once podocytes are lost, the remaining cells undergo hypertrophy to cover up the remaining surface, transmitting the damage and accelerating disease progression [33]. Furthermore, as described above, podocyte–endothelial cell crosstalk and interaction play an important role in the disruption of the GFB. Likely, variants in this type of cellular interplay may be partially responsible for some of the different subtypes of DN in terms of proteinuria and could be of a particular interest regarding individualized interventional therapeutics.

### 2.2. Altered Glomerular Hemodynamics and Proteinuria

Early T1D and T2D are associated with glomerular hyperfiltration, with subsequent reversal and decline in GFR over the progression of DN. The early hyperfiltration period is likely linked to the upregulation of sodium–glucose cotransporters (SGLT) in response to glycosuria, leading to enhanced sodium reabsorption in proximal tubules. Consequently, sodium chloride concentrations decline in filtrate reaching the macula densa, suppressing the tubulo-glomerular feedback mechanism. This, in turn, leads to the dilatation of glomerular afferent arterioles, with increased trans-glomerular pressure and glomerular hyperfiltration [34,35]. Figure 4 depicts the events that have been elucidated by the introduction of SGLT2 inhibitors (SGLT2i), revealing their efficacy in the reversion of afferent arteriolar vasodilation and hyperfiltration [36]. Structural glomerular remodeling in response to hyperfiltration develops over time, with an increase in glomerular size, capillary volume and filtration area [8,37], with a downstream hypertrophy of tubular segments. Renal hypertrophy is mediated by protein kinase C ꞵ1 [38] and conceivably predisposes to renal parenchymal hypoxia and hypoxia-related AKI [39,40]. Additional suggested mechanisms enhancing glomerular hyperfiltration in the diabetic kidney include increased hydraulic pressure gradient across the filtration barrier due to intense glucose, sodium and solute reabsorption in proximal tubules and increased renal interstitial pressure that may enhance sodium uptake in thin limbs [35].

Increased trans-glomerular hydraulic pressure induces hyperfiltration and augments the driving force leading to albumin leak across the filtration barrier. Indeed, measures that attenuate trans-glomerular pressure reduce albuminuria, often in parallel with reduced GFR. This was shown under acute settings first with NSAIDs [41], and later with captopril [42]. An acute reduction in transglomerular pressure and hyperfiltration has also been documented with the administration of ARBs [43] and SGLT2i [44], and the degree of reduced albuminuria was found to be proportional to the improved renal vascular resistance index following ARBs [45]. It is conceivable that this phenomenon is likely responsible for the long-term attenuation or prevention of albuminuria in large prospective clinical trials with ACEi, ARBs and SGLT2i [43,46,47]. Notably, the activation of hypoxia-inducible factors (HIF) also normalizes hyperfiltration [48], likely contributing to the attenuation of proteinuria.

As outlined below, hyperfiltration also can occur in response to high protein intake. It is estimated that eGFR rises by about 20 mL/min/1.73 m^2^ in normal individuals following a protein-enriched meal [49]. This response, associated with a surge in albuminuria, forms the basis for the low-protein diet recommended for diabetic patients with CKD, as discussed below.

Albuminuria reflects the net quantitative outcome of glomerular albumin leak and the capacity of tubular albumin re-uptake. Early diabetes is associated with hyperfiltration, related to activated RFR and nephron enlargement. Glomerular leak develops, initially with normalbuminuria, if it is surpassed by tubular capacity to reclaim filtered albumin. Microalbuminuria develops and progresses to macroalbuminuria as glomerular leak increases (ascribed to injured filtration barrier and increased trans-glomerular hydraulic pressure), combined with altered reclaiming of filtered albumin by proximal tubules. Creatinine is hardly affected at this stage thanks to fully activated RFR. However, with more advanced diabetic nephropathy, it rises with diminishing numbers of nephron units and exhaustion of RFR. Albuminuria may eventually decline as GFR and glomerular leak fall. 

Normalbuminuria appears in some 50% of patients with documented advanced diabetic glomerulopathy [50] and likely reflects sufficient proximal tubular capacity to remove filtered albumin and measures undertaken to diminish glomerular leak, such as ARBs, ACEi, SGLT2i and low protein intake.

### 2.3. Factors Affecting Trans-Glomerular Pressure and Glomerular Protein Leak: The Role of Declining Renal Functional Reserve

Protein ingestion is associated with an abrupt increment in GFR, a response that can be replicated by intravenous infusion of amino acids, such as glycine or alanine. This rise in GFR above baseline following protein load is referred to as renal functional reserve (RFR) [51]. It reflects the ability to enhance single-nephron GFR, likely by means of an increase in trans-glomerular pressure and/or ultrafiltration coefficient. RFR of intact nephrons may compensate, to a certain extent, for transient or permanent functional dropout of nephrons, maintaining near-normal levels of creatinine levels during a moderate decline in renal functional mass. Overall renal function may remain preserved at that range of moderate dropout of functioning nephrons, which can be acute and transient, or progressive and permanent, thanks to the activation of single-nephron functional reserve. In fact, clinical observations regarding acute changes in kidney function indicate that RFR may serve as a “shock absorber”, concealing “subclinical” AKI and recovery from AKI [52,53]. However, as shown in Figure 4, RFR diminishes as renal functional mass declines, likely since remnant nephrons activate their functional reserve to the maximum. As RFR diminished with advanced CKD, creatinine increments become steeper, and acute renal injury can no longer be masked by the buffering impact of RFR. This may explain the predisposition to AKI among diabetic and non-diabetic patients with advanced CKD, such as in the case of contrast nephropathy [40].

Of note, diabetes may compromise RFR irrespective of the decline in renal functional mass, as shown in a rat model of early streptozotocin-induced diabetes [54]. In humans with T1D, RFR was found to vary along disease progression. At the early hyperfiltrating stage, RFR likely declines, reflecting maximally hyperfiltrating glomeruli. As hyperfiltration regresses, RFR is restored, while it again diminishes in advanced stages of DN with macroalbuminuria, as remnant nephrons are maximally hyperfiltrating [55]. In patients with T2D and microalbuminuria, loss of RFR was more pronounced among individuals of African or Asian origin, and that has been linked to defective NO production or bioavailability [56]. RFR was also found to be inversely associated with markers of inflammation and with the activity of von Willebrand factor, suggesting a role for endothelial dysfunction [57].

It is tempting to assume that the association of protein load and the progression of DN and proteinuria are related to the activation of RFR during post-prandial hyperfiltration. While that may be the case in early diabetes with preserved RFR [58], long-term diabetes was found to abolish RFR response. However, fractional clearance of albumin profoundly increased in response to amino acid infusion, suggesting that albuminuria associated with protein load may not be associated with changes in renal hemodynamics only, and in advanced DN might rather reflect increased glomerular permselectivity to albumin [59].

### 2.4. Diabetic Tubulopathy—A Role in Albuminuria and Progressive DN

The renal tubule plays a pivotal role in renal handling of filtered albumin (Figure 3B). As previously reviewed in depth [60,61,62,63], a substantial fraction of filtered albumin is processed by tubular cells. Thus, albumin content in the final urine is much lower than that in the initial urine in Bowman’s capsule, and is near zero normally, as tubular retrieval capacity surpasses glomerular albumin leak.

Most filtered albumin is free and water-soluble, but a fraction is contained in clathrin-coated vesicles, formed by podocytes [63,64]. Both forms of albumin are retrieved principally by proximal tubular cells by means of endocytosis. Bound to cubilin and megalin, located at the brush border of proximal tubular cells [60,65], albumin is preferentially directed to lysosomes, with fragmentation and degradation to short peptides detected in the urine, and to amino acids transported to the bloodstream. Additionally, albumin, bound to a neonatal Fc receptor (FcRn), which also resides at the apical brush border, can pass intact through the tubular cell layer into the bloodstream (transcytosis), minimizing urinary losses as well as renal catabolism of albumin [63]. This is a high-capacity transport system, facilitated by a high affinity of albumin to FcRn in the acidic endosomal milieu, along with albumin release into the bloodstream at the higher pH in the extracellular environment [66]. Albumin may also be retrieved by paracellular mechanisms and in nephron segments other than the proximal tubule. A very small fraction is taken up by podocytes [64]. Tojo et al. [62,67] further suggest that albumin is also reabsorbed by distal convoluted tubules, and that the fraction of albumin reabsorbed in distal nephrons increases in the diabetic kidney as proximal tubular reuptake declines. However, to our knowledge, megalin and cubilin, the principal chaperones of retrieved albumin, are not expressed in distal nephron segments [65], and contamination by blood containing albumin could lead to possible erroneous interpretations of distal tubular albumin reclaiming during the micropuncture studies carried out by Tojo.

The amount of albumin reclaimed by proximal tubules by pinocytosis or transcytosisis is a matter of debate, attributed to a large dispute regarding the extent of glomerular leak. Human studies [68] as well as most experiments in rodents [62] indicate that the normal sieving coefficient of albumin at the filtration barrier (i.e., the ratio of filtered albumin to creatinine) is at the range of 0.0001–0.00008. By contrast, Russo et al. and Gekle [69,70] assessed a much higher sieving coefficient of 0.034 [62]. Based on in vivo two-photon technology and the determination of delayed peaks of labeled filtered albumin in the renal vein, they estimated that some 200–300 g of filtered albumin is normally retrieved daily, far beyond the capacity of the liver to synthesize albumin in compensation of renal degradation. They therefore concluded that the majority of filtered albumin is likely reclaimed intact by the transcellular pathway [60], and that based on studies of the fractional clearance of innate molecules with diverse sizes, compared to albumin, the megalin–cubilin low-capacity retrieval mechanism plays a minor role in tubular albumin retrieval [71,72]. Hypoalbuminemia was not prevented in nephritic megalin–cubilin KO mice [73], perhaps as reabsorbed albumin undergoes proteolysis. By contrast, hypoalbuminemia was corrected in FcRn KO mice [74], indicating its central role in transcytotic albumin retrieval in the nephrotic state. While the dispute regarding the relative importance of endocytosis/degradation, transcytosis and perhaps paracellular pathway in albumin reclaiming remains unsettled, most in vivo and ex vivo studies indicate that the amount of filtered albumin retrieved by tubular cells is more likely in the range of 3.5–7 g/day [66]. Additionally, albumin is fragmented and excreted into the urine as small degradation products, not detected by albumin assays [60]. It is estimated that about 1.3 grams of albumin normally undergo this pathway in the human kidney [75]. While the albumin retrieval system can easily cope with this modest degree of glomerular leak, a much larger burden of glomerular leak of albumin develops in diabetic and non-diabetic proteinuric diseases, with albuminuria appearing when glomerular leak exceeds the tubular retrieval capacity.

In that perspective, altered tubular albumin reclamation is an additional important factor leading to albuminuria in the diabetic kidney. Russo and colleagues [76] documented substantial attenuation of proximal tubular uptake of albumin in sreptozotocin-diabetic rats, using in vivo confocal microscopy while injecting fluorescence-labeled albumin. Importantly, this transport defect could be corrected by glycemic control, excluding a role for streptozotocin-related tubulotoxicity. They concluded that increased early albuminuria likely reflects the diminished capacity of tubular reclamation of albumin [76]. Additional studies in non-diabetic rats linked acute hyperglycemia to microalbuminuria, attributed to proximal tubulopathy with diminished expression of megalin and cubilin [77]. Indeed, an inverse association was found between megalin immunostaining in proximal tubules and the degree of albuminuria in human renal biopsies [78].

Renal tubular hypoxia may be an additional factor in the diminished capacity of albumin reclamation by the diabetic kidney. Low renal oxygenation has been recorded in experimental diabetes, preceding albuminuria [79] involving the cortex as well as the medulla [39]. It likely represents altered renal circulation, enhanced oxygen consumption for tubular transport and oxidative stress. Hypoxia likely predisposes the diabetic kidney to AKI, for example, in the setup of contrast nephropathy [40,80,81]. Upregulation of SGLT in the diabetic kidney reduces cortical oxygenation, while SGLT inhibition (hence, reduced transport load) ameliorates cortical oxygenation. On the other hand, it further intensifies medullary oxygen demand, conceivably due to enhanced solute delivery to the distal nephron, increasing sodium transport by medullary thick ascending limbs [82], likely predisposing to medullary hypoxic injury [83]. Indeed, diabetic patients hospitalized with AKI while on SGLT2i had increased urine and blood levels of NGAL, a marker of distal tubular injury, compared with diabetic patients with AKI who were not treated by SGLT2i, whereas KIM-1 levels (a biomarker of proximal tubular injury) were comparable [84]. With all that in mind, it is tempting to assume that diminished albumin re-uptake and amino acid transport in diabetic proximal tubular cells (additional energy-demanding processes) might reflect hypoxic cell dysfunction. In line with this hypothesis is the attenuation of albuminuria attributed to improved tubular albumin claiming with the administration of RAS blockers or SGLT2i. These classes of medications attenuate proximal tubular oxygen consumption, either directly (SGLT2i) or indirectly (RAS blockers as well as SGLT2i, via reduced trans-glomerular pressure, GFR and solute delivery for tubular transport). In the same fashion, HIF activation, through attenuating hyperfiltration, reduces tubular electrolyte load and improves renal oxygenation. This in turn reduces oxidative stress and tubular injury and likely contributes to improved tubular albumin uptake [48]. Notably, SGLT2i may also trans-activate tubular SIRT-1 and HIF, promoting autophagy and attenuating tubulointerstitial inflammation [85].

### 2.5. Albuminuria Accelerates the Progression of DN

Remuzzi et al. proposed that tubular handling of filtered albumin is, to a large extent, responsible for tubular damage, inflammation and the progression of CKD in proteinuric patients [86]. As outlined below, measures that reduce proteinuria in diabetic and non-diabetic renal diseases also attenuate the progression of CKD. It is difficult to distinguish between the direct nephroprotective impact of reducing albuminuria from that of other mechanisms, such as the control of hypertension, restoration of glomerular hemodynamics attenuating hyperfiltration or the improvement of renal parenchymal oxygenation. Yet, as thoroughly discussed elsewhere [87], clinical trials in secondary hypothesis generating analysis suggested that the impact of reducing proteinuria was independent from that of controlling blood pressure [88], and that early lowering of proteinuria predicted subsequent preservation of renal function [89]. By means of tubular protection, protective interventions could improve tubular reclamation of albumin and reduce albuminuria. However, it conceivably works principally the other way, i.e., that reduced filtered albumin protects tubular cells.

The mechanisms by which albuminuria generates tubulointerstitial disease were first revealed by pioneering studies, conducted by Schreiner and colleagues [90,91,92], leading to the concept of albuminuria-induced liponephrotoxicity. They hypothesized that fatty acids accumulate in proximal tubular cells following endocytosis of fatty acid-loaded albumin, as well as by specific fatty acid transporters, discussed below. Heavy albuminuria leads to the synthesis and accumulation of triglycerides [92]. Lipid-loaded tubular cells undergo vacuolization and ultimately take the form of “oval fat bodies”, a characteristic finding of urinalysis in patients with nephritic syndrome [93]. More importantly, accumulated fatty acids dissolved in endocytosed albumin are potent chemoattractants [90] that trigger macrophages in vivo. A non-polar lipid released from proximal tubular segments loaded by albumin in vivo or in vitro showed similar effects, while tubular segments incubated with fatty acid-depleted albumin did not induce macrophage chemoattractivity [91]. Using a model of metabolic syndrome with hereditary hypertriglyceridemia in rats, Markova et al. [94] reported renal cortical accumulation of neutral lipids and triglycerides, but not of cholesterol, associated with increased urinary excretion of inflammatory mediators (MCP-1, IL-6 and IL-8) and decreased urinary secretion of epidermal growth factor (EGF) that preceded the appearance of microalbuminuria. Lipid accumulation was demonstrated in proximal tubular cells, but the mode of accumulation, presumably via endocytosis, has not been addressed. Indeed, liponephrotoxicity may not be solely mediated by the uptake of fatty acids dissolved in reclaimed albumin. Khan et al. [95] proposed that fatty acids may be directly taken up into proximal tubular cells in diabetics by fatty acid transport protein-2 (FATP2), located at the luminal brush border, likely irrespective of albumin retrieval. They found that mutated diabetic mice prone to CKD and lacking FATP2 showed normalization of GFR, reduced albuminuria, improved kidney histopathology and longer life span compared with diabetic animals expressing FATP2.

Thus, albuminuria and liponephrotoxicity might be bi-directional, i.e., the prevention of albuminuria may halt proximal lipid-mediated tubulopathy, and the attenuation of lipid accumulation in proximal tubular cells may restore tubular function, improving tubular albumin re-uptake and reducing albuminuria. It has been proposed that supplementation of long-chain omega-3 polyunsaturated fatty acids may affect the composition of lipids retained in proximal tubular cells and ameliorate albuminuria [96]. Furthermore, restoration of mitochondrial function and integrity by mitochondrial-targeted antioxidant SS-31 attenuated renal injury, oxidative stress and inflammation and diminished albuminuria in streptozotocin-diabetic rats [97], in part, likely by the restoration of mitochondrial beta-oxidation [98].

In addition to liponephrotoxicity, tubulointerstital inflammation in the diabetic kidney is likely also triggered by hyperglycemia, with documented induction of pro-inflammatory and pro-fibrotic pathways, leading to progressive diabetic CKD [61,99]. Thus, the control of hyperglycemia, in concert with abolishment of hyperfiltration and albuminuria, is essential for the long-term integrity of the diabetic kidney. Hypoxia is another possible component in the pathogenesis of tubular damage in the diabetic albuminuric patient. Albumin transport and degradation in proximal tubular cells are energy consuming, and may further intensify renal cortical hypoxia, discussed above, caused by enhanced coupled reabsorption of glucose and sodium. Indeed, urinary excretion of kidney injury molecule (KIM)-1, a marker of proximal tubular injury, significantly declined in patients on empagliflozin [100], presumably reflecting improved cortical oxygenation.

Notably, liponephrotoxicity may also involve podocytes [101], possibly enhancing glomerular leak. This is not surprising considering that a fraction of the filtered albumin is up taken by podocytes as well [64].

To conclude, as illustrated in Figure 1, albuminuria-associated tubulotoxicity is an inherent component in the progression of DN. Together with evolving glomerulopathy, it leads to the development of advanced CKD, a complication preventable by strategies detailed below, that in concert diminish glomerular hyperfiltration and albuminuria.

### 2.6. Integrated Glomerulo-Tubular Adjustments Affecting Proteinuria

The degree of proteinuria is substantially affected by the interactions of the various factors discussed above, which may counteract each other or act in concert. As outlined by Tonneijck et al. [35] and illustrated in Figure 4, at early stages of diabetic kidney disease, glomerular structural damage is expected to be low, and albuminuria may not be present despite hyperfiltration, as long as tubular reclamation capacity of filtered albumin is preserved. With more advanced glomerulopathy, proteinuria will develop, especially if hyperfiltration is still maintained. As DN progresses, hyperfiltration is reversed and a decline in GFR might reduce proteinuria, if not for the progressive damage of the filtration barrier and the declining capacity of tubular re-uptake of albumin. At more advanced stages of nephropathy, single-nephron GFR in remnant nephron units may still be high, contributing to albuminuria, with a possible additional impact of rapid solute movement and shortened transit time along the malfunctioning tubular cells, which may further interfere with reclamation of albumin. Yet, with far-advanced CKD, with eGFR below 15–20 mL/min/1.73 m^2^, reduced filtration will likely become dominant, reducing the absolute magnitude of proteinuria [35]. This hypothetical scheme cannot be tested in clinical practice but might be evaluated experimentally with the combined use of technologies such as confocal video microscopy [76], micropuncture studies and the assessment of morphologic counterparts. Proteomics and genomics may further elucidate a role for diverse expression and functionality of various candidate players in the generation and elimination of albuminuria, such as megalin and cubilin, vasoactive compounds and their receptors or participants in the molecular cascade leading to inflammatory response and renal parenchymal fibrosis [102,103,104]. Noteworthy, so far, such evaluation has excluded a role for single nucleotide polymorphism among components of the slit diaphragm [105]. However, potential candidates may be genes involved in diabetic tubulopathy (such as MCP-1), in tubulointerstitial inflammation and fibrosis (YKL-40 and the like) or engaged in tubular cell repair or proximal tubular reabsorptive capacity [106,107,108]. Heterogeneous bioavailability of such elements may help explain enigmas such as non-albuminuric DN or the absence of diabetic kidney disease despite long-lasting uncontrolled diabetes with overt non-renal organ damage.

## 3. Clinical Perspectives

### 3.1. Attenuating Proteinuria Is Nephroprotective

Large clinical data repeatedly show that interventions that reduce trans-glomerular pressure and proteinuria improve renal functional outcome in diabetics and in non-diabetic renal diseases, delaying the decline in GFR and the need for renal replacement therapy [87,109]. This indicates that measures aimed at reducing proteinuria translate into protection from decline in renal function in albuminuric patients with and without diabetes with the interventions tested thus far in clinical trials. It is possible that reduced proteinuria could be regarded as a surrogate end point for clinical trials in albuminuric patients if more interventions reveal a consistent association [110]. However, there have been exceptions recorded. Renal protection in non-albuminuric diabetics may still be carried out via reduced tubular load of albumin (under conditions where tubular reclaiming of albumin fully compensates for enhanced glomerular leak), or may emphasize the impact of reduced glomerular changes related to hyperfiltration and barotrauma per se.

A major early report linking reduced proteinuria and renoprotection in T2D patients is the RENAAL study. A 26% decline in proteinuria was noted within 3 months in patients given losartan, further declining by over 40% within 3 years, with a profound improvement of renal survival [111]. Comparable findings were reported in a parallel study, with a significant reduction in adverse renal outcomes with irbesartan [112]. Other ARBs, ACEi or their combination were equally effective in attenuating proteinuria and the progression of renal dysfunction [46], underscoring a class effect with a common physiologic nephroprotective mode of action. Finerenone, a non-steroidal mineralocorticoid antagonist, was also found to rapidly reduce proteinuria in diabetic patients already on RAS blockers [113], with long-term renal protection, as well [114].

The introduction of SGLT2i revolutionized the clinical management of T2D, markedly improving overall and cardiovascular outcomes. These agents are also renal-protective in both diabetic and non-diabetic patients with moderate and severe renal failure and exert a positive effect on proteinuria [115]. Indeed, marked renal protection and a reduction in proteinuria were evident in several cardiovascular outcome studies in over 35,000 T2D patients [47,116,117]. DECLARE, being the largest cardiovascular-outcome-trial with SGLT2 inhibitors [116], included over 10,000 T2D patients with cardiovascular risk factors but without established cardiovascular disease, with relatively normal kidney function tests (48% with eGFR > 90 mL/min/1.73 m^2^ and 45% with eGFR 60–90 mL/min/1.73 m^2^, while only 7% were with eGFR < 60 mL/min/1.73 m^2^). About 30% of these patients had microalbuminuria and 7% had proteinuria. The rate of renal deterioration was comparable in the three groups of patients stratified by their initial eGFR. However, renal deterioration was maximal in those with the combination of renal failure and proteinuria [118]. Dapagliflozin prevented deterioration in eGFR to the same extent in the three study groups, unrelated to baseline eGFR, or to nonexistence or existence of micro or macro albuminuria.

As compared to SGLT2i agents that exert both renoprotective and anti-proteinuric effects, activation of the incretin axis in T2D patients was found to attenuate proteinuria, but without slowing the decline in kidney function. Indeed, clinical trials evaluating the effect of DPP4 inhibitors revealed the attenuation of albuminuria without a parallel effect on renal outcome [119,120,121]. Similarly, GLP1 agonists reduce proteinuria but are not proven to attenuate renal functional deterioration [122,123,124]. The ongoing FLOW trial (NN9535-4321) using Semaglutide is further testing this hypothesis [125].

Taken together, these outcomes of large prospective interventional studies in patients with T2D outlined above reveal that the attenuation of proteinuria with the use of RAS blockade or with SGLT2i is accompanied by a slowing of the progression of renal functional impairment. This favors the possible role of proteinuria in the progression of DN. However, treatment with DPP4i or with GLP1 agonists attenuates proteinuria but probably has no or little impact on the progression of renal functional impairment. Furthermore, RAS inhibition and the use of SGLT2i apparently may attenuate the progression of DN even in the absence of proteinuria. Thus, with the current clinical data, we cannot unequivocally dissect the independent impact of attenuating proteinuria on renal salvage from renal protection provided by an overall control of metabolic derangements, systemic hypertension and glomerular hypertension and hyperfiltration. Of note, however, the findings of a recent study reporting renoprotective properties of Dapagliflozin unrelated to diabetes in patients with focal segmental glomerulosclerosis (FSGS) strengthen the potential independent impact of reducing proteinuria on the attenuation of progressive diabetic nephropathy [126]

Irrespective of whether the attenuation of proteinuria contributes to the renoprotective mechanism of SGLT2i, we suggest that this group of drugs should be considered for every patient with T2D who does not have a clear contraindication. It is anticipated to be effective and safe for the entire spectrum of baseline eGFR (exept for far-advanced CKD, with or without microalbuminuria). The number needed to treat, however, is expected to be much higher in those with normal kidney function and without albuminuria.

DPP4 antagonists and GLP1 analogues do not immediately affect glomerular hemodynamics. Therefore, we propose that their impact on reducing albuminuria might be mediated indirectly through the gradual improved glycemic and metabolic control (with restoration of inhibited TGF, reduction in trans-glomerular pressure and ameliorated cortical oxygenation), with a possibly better tubular handling of filtered albumin. A gradual mild decline in GFR within 12 weeks with GLP1 analogues, noted in the SUSTAIN series [127], is in line with this notion, probably reflecting reversal of hyperfiltration. Their lack of proven effect, so far, on long-term renal outcomes, as opposed to RAS blockade of SGLT2i may be explained by a more gradual impact on the restoration of glomerular hemodynamics, with an anticipated longer time required for the detection of an effect, or additional mechanisms that might be involved by the other therapeutic modalities, such as the attenuation of hypoxia/HIF-mediated progressive injury and fibrosis.

Table 1 illustrates the comparison of all the modalities discussed above in respect totheir impact on proteinuria and renal protection.

### 3.2. When to Initiate Anti-Proteinuric Treatment?

The presence of advanced renal failure among diabetics in the absence of proteinuria has long been documented [4,128], and its pathogenesis remains enigmatic. In some cases, causes other than diabetic changes may be involved, such as hypertensive nephropathy, unrecovered AKI or renovascular disease. A recent study by Sasaki et al. [50] provides new perspectives regarding this mystery, looking at renal diabetic injury detected during autopsy of 108 aged diabetic Japanese patients and their degree of preceding albuminuria. Expectedly, the degree of advanced diabetic glomerulopathy (stages 2 and 3) was proportional to the magnitude of proteinuria. The striking unexpected finding, however, was that about half of the non-albuminuric patients had advanced diabetic glomerulopathy, indicating that the absence of albuminuria does not rule out the presence of DN. We propose that this might be explained by a substantial decline in filtered albumin, for instance, with an advanced decline in trans-glomerular pressure and GFR. Indeed, some 40% of these patients were on RAS-blocking agents. Moreover, pathological analysis did not address the degree of tubulointerstitial changes, and plausibly preserved efficient tubular handling of filtered albumin could conceal glomerular albuminuria. Further comparable clinical and experimental studies are required to evaluate these possibilities, but for the time being, a breakthrough conclusion that might be adopted from Sasaki’s findings is to initiate early treatment with RAS blockers or SGLT2i in diabetic patients, even before microalbuminuria develops. Noninvasive future technologies may help to differ between patients with “silent” diabetic kidney disease that deserve treatment and those with intact kidneys who do not as yet [129].

### 3.3. The Protein Restriction Controversy

For some 75 years, low protein intake has been a cornerstone of the management of CKD, with an arguable favorable impact on retarding the progression of renal dysfunction [130]. The rationale for this approach has been the prevention of glomerular hyperfiltration and suppressing the generation of uremic toxins [131,132]. Indeed, protein restriction attenuated proteinuria and renal lesions in experimental animal studies [133] and slowed the progression of CKD in humans in some but not all studies [134]. By contrast, a high-protein diet markedly increases post-prandial GFR in normal individuals [49] and moderately increases fasting GFR by 4 mL/min/1.73m^2^ [135]. Furthermore, high protein intake accentuates albuminuria in nephrotic patients [136] and accelerates the progression of CKD in humans—as an example, T1D patients with microalbuminuria fed a low-protein diet for one year presented with diminished albuminuria and restored RFR and their decline in kidney function had been attenuated, compared with patients fed normal-protein diets who showed opposite outcomes [134]. In a short-term study, Kaysten et al. [137] followed nine non-diabetic nephrotic patients intermittently fed high- and low-protein diets. They found that renal albumin clearance and fractional renal albumin clearance indeed declined, in excess of any reduction in creatinine clearance, when the patients were fed low-protein diets. Interestingly, the impact of renal albumin degradation/loss surpassed that of albumin synthesis, as despite enhanced albumin synthesis with a high-protein diet, plasma albumin and total albumin mass declined [137].

Thus, protein restriction seems reasonable in the attenuation of renal hyperfiltration and proteinuria, with a potential long-term favorable impact on renal survival. However, protein restriction may inhibit hepatic albumin synthesis [138], which might be further suppressed by inflammatory signals [139], characterizing DN [140]. Furthermore, diabetes [141] as well as CKD [142] are associated with sarcopenia, and the likelihood of sarcopenia among diabetics increases 2.54-fold in the presence of DN [141]. Renal protein wasting, discussed above, possibly contributes to sarcopenia, and intuitively one would favor enriched protein intake rather than a low-protein diet specifically in this population.

A recent article by Oosterwijk et al. [143] forms a breakthrough in the dilemma regarding protein restriction in patients with DN and CKD. In this observational prospective study, the long-term impact of lifestyle on renal outcome was evaluated. The study included 382 adults with T2D (mean age 63 years), obese (mean BMI 32.8 kg/m^2^) and predominantly male (59%) patients. Daily protein intake in outpatient settings was assessed by 24 h urea excretion. Surprisingly, dietary protein intake was inversely associated with renal functional deterioration. Patients with a protein intake >163 g/day had a decreased hazard for renal function deterioration (HR 0.42), whereas patients with an intake <92 g/day had an increased hazard for renal functional deterioration (HR 1.44) over a median follow-up period of 6 years [143].

There are several possible explanations for these unexpected findings which contradict our current clinical practice. The first one is that whereas the original studies showing nephroprotection by protein restriction addressed naïve diabetic renal disease without pharmaceutical interventions, 72% of the patients included in Oosterwijk’s study were on ARBs and ACEi, while others were possibly on SGLT2i. Thus, conceivably, these medications prevented glomerular hyperfiltration and enhanced proteinuria invoked by high protein intake, preventing consequent glomerular sclerosis, hyalinosis and tubulointerstitial disease [144]. This possibility is in line with studies by Kaysen et al., illustrating a reversal of hyperfiltration mediated by high protein intake in nephrotic rats treated with ACEi [145]. Secondly, as illustrated in Figure 4, as RFR declines with advanced CKD, the ability to mount an increase in GFR following protein load diminishes, again attenuating harmful glomerular hyperfiltration. Finally, differences in other food constituents, such as fat composition, could blunt the impact of albumin restriction on proteinuria [146].

Thus, the consensus regarding an obligatory protein restriction in patients with DN deserves reassessment. Additional large-scale prospective studies are of course needed to validate Oosterwijk’s findings [143]. Our proposed explanations regarding the contradicting current and historical outcomes may be tested by looking at the impact of acute protein load on the degree of proteinuria with and without RASi and/or SGLT2i at different stages of RFR status along the full range of GFR.

## 4. Conclusions

Albuminuria in patients with DN reflects on one hand an altered GFB and glomerular hemodynamics, as well as the activation of renal functional reserve upon the introduction of a protein load, all leading to the glomerular leak of albumin. On the other hand, albuminuria may reflect defective renal tubular capacity of reabsorption and breakdown of filtered albumin. Declining GFR, loss of RFR and diminishing tubular reabsorptive capacity along the progression of DN likely have profound and diverse impacts on the magnitude of albuminuria. A better understanding of the fate of filtered albumin probably explains some discrepancies, such as the common phenomenon of glomerulopathy sans proteinuria [4,50,128]. The individual impact of the many variables affecting proteinuria and their associations, including glomerulopathy, hyperfiltration, RFR and protein intake, and tubular albumin reclaiming capacity are only partially understood. Albuminuria, even if controlled by efficient tubular uptake, harbors the risk for the development of tubulointerstitial damage with the progression to CKD. With that in mind, initiation of RASi or SGLT2i in diabetics prior to the appearance of microalbuminuria should likely be adopted. Furthermore, reassessment of protein restriction is needed in patients with renal functional impairment who are on medications that prevent glomerular hyperfiltration.

## Figures and Tables

**Figure 1 cells-11-02917-f001:**
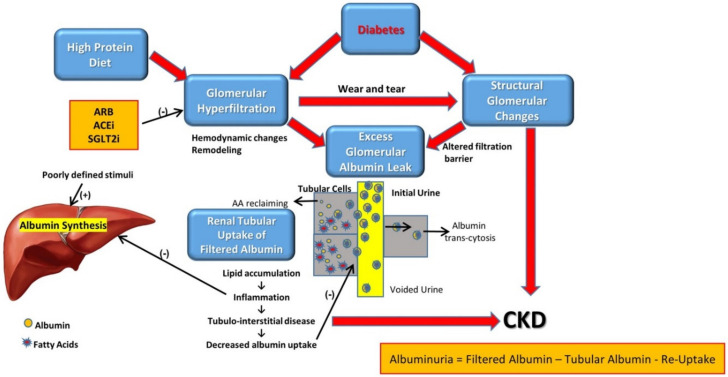
Scheme of albumin fate within the kidney and its association with the pathogenesis of diabetic nephropathy.

**Figure 2 cells-11-02917-f002:**
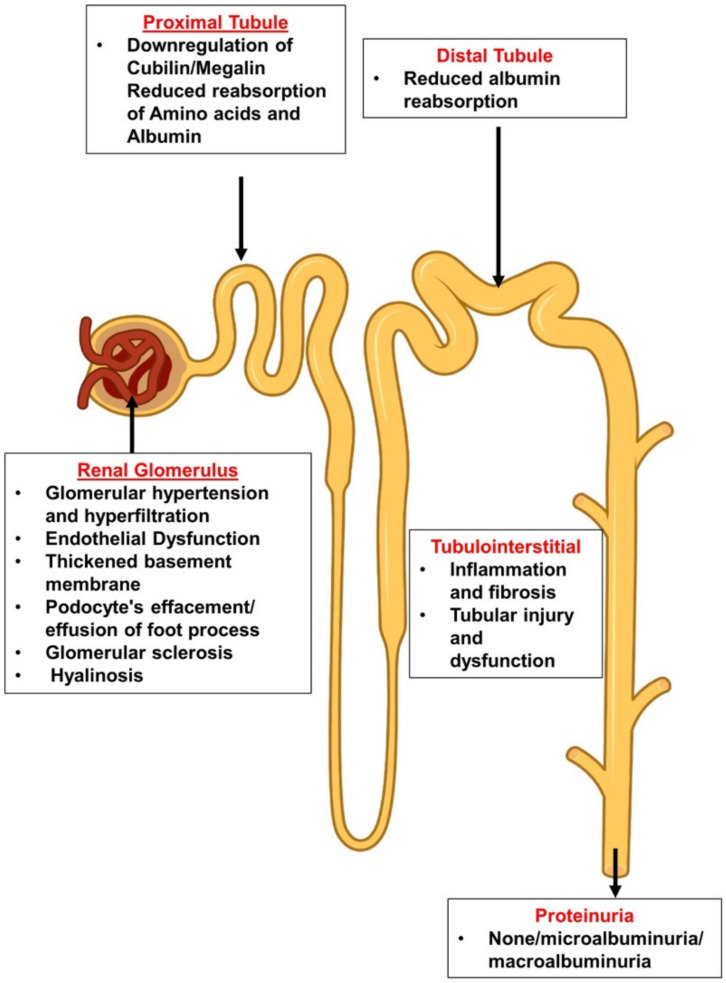
Glomerular and tubular deleterious alterations in the diabetic nephron.

**Figure 3 cells-11-02917-f003:**
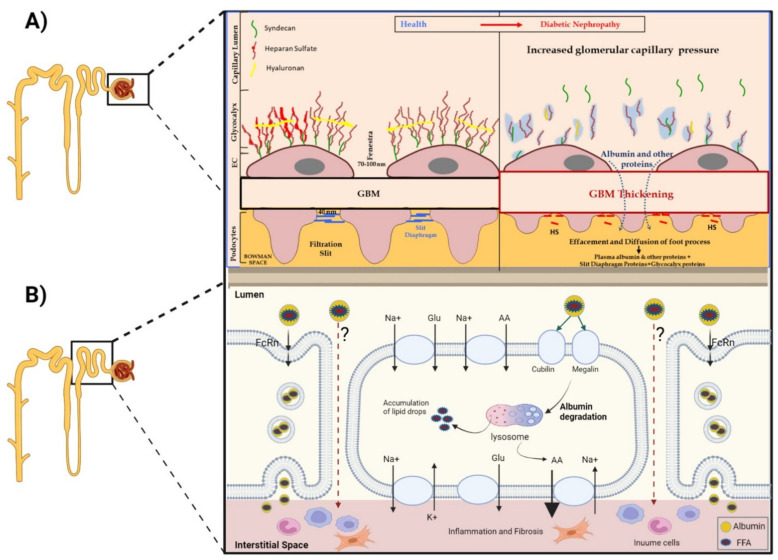
(**A**) Impact of DM on the ultrastructure of the various layers constituting the glomerular filtration barrier (GFB). (**B**) Pathways of tubular reabsorption of filtered albumin.

**Figure 4 cells-11-02917-f004:**
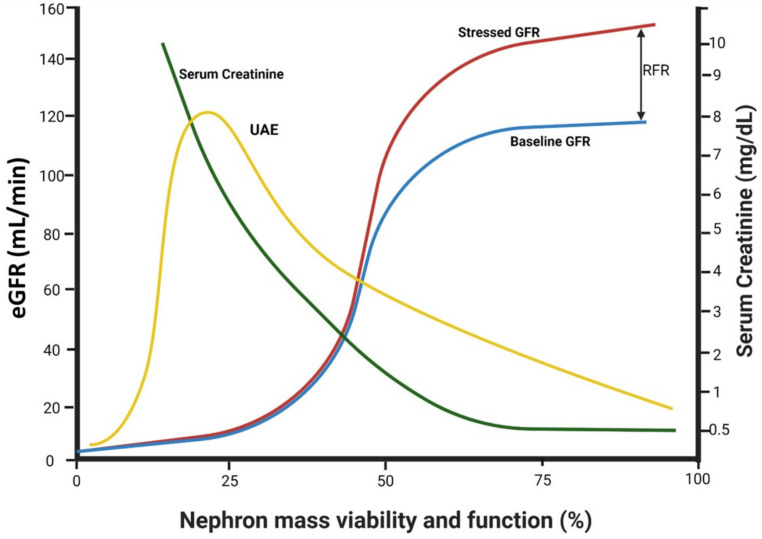
The impact of declining functional renal mass on GFR, RFR and albuminuria.

**Table 1 cells-11-02917-t001:** Pharmaceutical interventions in the management of diabetic kidney disease.

Drug Class	Impact on Proteinuria	Acute Effects on Kidney Function	Long-Term Impact on Kidney Function	Mechanisms Leading to Reduced Proteinuria	Mechanisms of Renal Protection
ACE inhibitors	Reduce proteinuria	Reduce GFR	Renoprotective	- Reduce glomerular hypertension via efferent arteriolar vasodilation - Improved metabolic control attenuates structural glomerular changes - Improved tubular protein re-uptake?	- Reducing proteinuria leads to attenuated tubulointerstitial disease - Blocking Ang II/aldosterone-mediated inflammation, tubular apoptosis and fibrosis
ARBs	Reduce proteinuria	Reduce GFR	Renoprotective	Reduce glomerular hypertension via efferent arteriolar vasodilation	- Reducing proteinuria leads to attenuated tubulointerstitial disease - Blocking Ang II/aldosterone-mediated inflammation, tubular apoptosis and fibrosis
SGLT2 inhibitors	Reduce proteinuria	Reduce GFR	Renoprotective	Immediate reduction in glomerular hypertension via restoration of tubulo-glomerular feedback (TGF)	- Improved metabolic control attenuates structural glomerular injury - Reducing proteinuria leads to attenuated tubulointerstitial disease - Possible additional anti-fibrotic properties
DPP4 antagonists	Reduce proteinuria	None	Probably no impact	Restoration of metabolic derangements leading to gradual reduction in glomerular hypertension via restoration of TGF with afferent arteriolar vasoconstriction afferent arteriolar vasoconstriction	- Improved metabolic control may attenuate structural glomerular injury - Reducing proteinuria may lead to attenuated tubulointerstitial disease
GLP1 analogues	Reduce proteinuria	None Mildly reduced GFR by 12 weeks	Currently unknown	Restoration of metabolic derangements leading to gradual reduction in glomerular hypertension via restoration of TGF with afferent arteriolar vasoconstriction afferent arteriolar vasoconstriction	- Improved metabolic control may attenuate structural glomerular injury - Reducing proteinuria may lead to attenuated tubulointerstitial disease

## Data Availability

Not applicable.

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
