# Peer review of "Diabetic Proteinuria Revisited: Updated Physiologic Perspectives"

_cells, 2022, doi:10.3390/cells11182917_

Round 1

Reviewer 1 Report

In this manuscript, the authors concluded the factors leading to albuminuria, the role of albuminuria in the progression of DN, the controversy about protein intake, and the anti-proteinuric treatment. However, the logic of the article is confusing.

1. The pathological mechanisms of proteinuria include deformed filtration barrier, altered glomerular hemodynamics and hyperfiltration, and the inability of renal tubules. However, the author inserted “plausible implications regarding proteinuria” in the part of mechanism description.

2. This part “Interactive factors affecting proteinuria” can be included in the mechanism description part.

3. Supplement the summary of current clinical drugs for proteinuria, their functional mechanism and therapeutic effect.

4. Supplement one or more mechanism scheme leading to proteinuria, especially the more clearly studied molecules and signaling pathways involved in it.

Author Response

Dear Editor

We would like to thank you and the reviewers for the contributing comments that, we believe, led to a substantial improvement of our manuscript.

We believe that you will find the enclosed revised version clearer and easier to follow, and hope you find it suitable for publication in Cells.

Sincerely, yours

Samuel N Heyman

In this manuscript, the authors concluded the factors leading to albuminuria, the role of albuminuria in the progression of DN, the controversy about protein intake, and the anti-proteinuric treatment. However, the logic of the article is confusing.

The pathological mechanisms of proteinuria include deformed filtration barrier, altered glomerular hemodynamics and hyperfiltration, and the inability of renal tubules. However, the author inserted “renal functional reserve: plausible implications regarding proteinuria” in the part of mechanism description.

We disagree with your comment as we believe this section is a reasonable continuation of the previous one dealing with the pathogenesis of glomerular protein leak. In this section we address renal functional reserve and protein intake, two factors that markedly govern trans-glomerular pressure and proteinuria. However, your comment correctly indicates that the title of this paragraph is misleading and requires modification. In the new version it has been changed into " Factors affecting trans-glomerular pressure and glomerular protein leak: the role of declining renal functional reserve". With this modification, we believe the evolution of the narrative from the previous section is smoother and the order or the manuscript makes sense.

This part “Interactive factors affecting proteinuria” can be included in the mechanism description part.

Again your remark underscores our failure to clearly illustrate the structure of the review. We now segregated the manuscript into four main sections: Introduction; Pathophysiology; Clinical perspectives; and Conclusions. The middle two sections are now divided, each one to several chapters. The protein restriction chapter was moved to the Clinical Perspectives section. The chapter you comment upon is the last one in the Pathophysiology section, as it deals with the integrated glomerulo-tubular function affecting proteinuria. We modified its title into " Integrated glomerolo-tubular adjustments affecting proteinuria", underscoring its correct placement at the conclusion of the Pathophysiology section.   

Supplement the summary of current clinical drugs for proteinuria, their functional mechanism and therapeutic effect.

A Table 1 in now added, summarizing clinical trials with the different drug classes and showing presumed functional mechanusms attenuating proteinuria

  1. Supplement one or more mechanism scheme leading to proteinuria, especially the more clearly studied molecules and signaling pathways involved in it.

In the revised manuscript we incorporated two new schemes showing structural and functional alterations participating in proteinuria (Figures 2 and 3 in the revised submission) 

Thank you very much

Reviewer 2 Report

Heyman et al. review the mechanisms of albuminuria and summarize the potential treatment options including SGLT2-inhibitors and the renin-angiotensin-aldosterone system blockers in diabetic nephropathy in their review article. The topic is important and interesting; the text is well-written and easy to follow. 

1. Fig. 1 is a good summary figure; however, the authors discuss more detailed mechanisms in the text compared to the Figure. The mechanisms and changes in different cell types leading to a deformed filtration barrier in DN would improve the understanding of the text.

2. It could be helpful for the readers if the drug targets and effects were depicted in a Figure (SGLT2i, RAS blockers vs. DPP4 antagonists and GLP1 analogs). 

3. The explanation of abbreviations is missing in the legend of Fig. 1.

4. It would improve the clarity of text if the names and main results of the mentioned clinical trials were visualized in the form of a Table also.

Author Response

Dear Editor

We would like to thank you and the reviewers for the contributing comments that, we believe, led to a substantial improvement of our manuscript.

We believe that you will find the enclosed revised version clearer and easier to follow, and hope you find it suitable for publication in Cells.

Sincerely, yours

Samuel N Heyman

Heyman et al. review the mechanisms of albuminuria and summarize the potential treatment options including SGLT2-inhibitors and the renin-angiotensin-aldosterone system blockers in diabetic nephropathy in their review article. The topic is important and interesting, the text is well-written and easy to follow

Fig. 1 is a good summary figure; however, the authors discuss more detailed mechanisms in the text compared to the Figure. The mechanisms and changes in different cell types leading to a deformed filtration barrier in DN would improve the understanding of the text.

Thank you for this comment. In the revised version we now add two new figures. The first one (Figure 3 in the revised submission) provides a better view at the glomerular and tubulo-  interstitial levels, providing more details regarding structure and cellular physiology linked to albuminuria, that could not be shown in detail in Figure 1. The new Figure 2 provides an additional simplistic overview of renal albumin handling and fate in the diabetic kidney.     

It could be helpful for the readers if the drug targets and effects were depicted in a Figure (SGLT2i, RAS blockers vs. DPP4 antagonists and GLP1 analogs). 

This is also a good suggestion. Figure 1 already depicts the site of action of SGLT2i and RAS inhibiton. In an additional new Table 1 we now address the comparisons of these medications and DPP4i / GLP1 analogues regarding their impact on proteinuria and renal function, addressing also their their physiologic mechanism of action regarding the fate of proteinuria.  In the text we now also depict the plausible renoprotective mechanism of GLP1 analogues that involves restoration of glomerular hemodynamics, with a by-product of decreasing proteinuria

The explanation of abbreviations is missing in the legend of Fig. 1.

Abbreviations have been added to the legend

It would improve the clarity of text if the names and main results of the mentioned clinical trials were visualized in the form of a Table also.

As stated above, in the added new Table 1, the outcome of clinical trials with the different drug classes and their presumed modes of action attenuating proteinuria are outlined. For the sake of simplicity we decided to address drug classes and to avoid a detailed list of the clinical trials, detailed in the text.

Round 2

Reviewer 1 Report

 Accept in present form.

Reviewer 2 Report

The authors answered my questions and modified their MS accordingly. The two new figures and the new table improved the understanding of the MS markedly. I recommend accepting their modified MS for publication in Cells.